



**Hydrologic modeling of a Himalayan mountain basin by using the SWAT model**
**Sharad K. Jain[1]\*, Sanjay K. Jain[1], Neha Jain[1] and Chong-Yu Xu[2]**
[1]National Institute of Hydrology, Roorkee 247667, India
[2] University of Oslo, Oslo, Norway
\*Corresponding Author: Email s_k_jain@yahoo.com
**ABSTRACT**
A large population depends on runoff from Himalayan rivers which have high hydropower
potential; floods in these rivers are also frequent. Current understanding of hydrologic response
mechanism of these rivers and impact of climate change is inadequate due to limited studies. This
paper presents results of modeling to understand the hydrologic response and compute the water
balance components of a Himalayan river basin in India viz. Ganga up to Devprayag. Soil and
Water Assessment Tool (SWAT) model was applied for simulation of the snow/rainfed catchment.
SWAT was calibrated with daily streamflow data for 1992-98 and validated with data for 1999-
2005.Manual calibration was carried out to determine model parameters and quantify uncertainty.
Results indicate good simulation of streamflow; main contribution to water yield is from lateral and
ground water flow. Water yield and ET for the catchments varies between 43-46 % and 57-58% of
precipitation, respectively. The contribution of snowmelt to lateral runoff for Ganga River ranged
between 13-20%. More attention is needed to strengthen spatial and temporal hydrometeorological
database for the study basins for improved modeling.
Keywords: Hydrological modeling, SWAT, Western Himalaya, calibration
**1 INTRODUCTION**
Many rivers, springs and lakes in the mountain regions are fed by significant contribution runoff
from snow and glacier melt. The headwater catchments of most of the rivers in the Himalayan
region such as the Ganga, the Indus and the Brahamaputra, lie in the snow covered areas. Snowfall
is temporarily stored in high hills and the melt water reaches the river later in the hot season. Snow
and glacier runoff are vital in making big Himalayan Rivers perennial whereas the rainfall
contribution during the monsoon season is important for high flow volumes in rivers. Snow
accumulates in the Himalayas generally from November to March, while melt season spans the
months April to September. Snowmelt is the predominant component of runoff in mountains in
April to June months and it forms a significant constituent of streamflows during July - September.

High spatial and temporal variability in hydro-meteorological conditions in mountainous
environments requires spatial models that are physically realistic and computationally efficient
(Liston and Elder, 2006b). Among the models developed to simulate hydrological response of
mountainous basins, the most common approach followed for distributed snowmelt modeling in the
absence of detailed measured data is to subdivide the basin into zones based upon elevation,
allowing the model to discretize the snowmelt process based on watershed topography (Hartman et





al., 1999; Li et al., 2013, 2015, 2016). The list of models developed for modeling response of a
catchment subject to solid and liquid precipitation includes commercial software such as Mike-SHE
(http://mikebydhi.com) and the public domain models such as the SWAT model (Neitsch, 2002), the
Xinanjiang Model (Zhao et al., 1995)and the HBV model (Bergstrom, 1992). An obvious advantage
of the public-domain models is the saving in cost and ease in sharing model set-ups. SWAT is a
public-domain model that has been used extensively. A user-friendly interface to set-up the model
in a GIS framework, detailed user's manual and a large user base are the main reasons for a number
of applications of the SWAT model.

SWAT is a semi-distributed, continuous watershed modelling system, which simulates
different hydrologic responses using process based equations. Most of the applications of the
SWAT model have used daily or monthly time steps for simulation. Obviously, it has been
comparatively easy to obtain higher values of Nash-Sutcliffe Efficiency (NSE) for monthly data
than for the daily data. Further, SWAT model has been successfully applied to catchments with size
of a few sq. km to thousands of sq. km. For example, Spruill et al., 2000 applied the SWAT model
to simulate daily streamflows in a watershed in Kentucky covering an area of 5.5 km$^2$ whereas
Zhang et al., 2008 used it to simulate monthly runoff of a mountainous river basin in China
covering area of 114,345 km$^2$. Some more recent applications of SWAT for rainfall-runoff
modeling are those by Jain et al., 2010, Shawul et al., 2013, Kushwaha and Jain, 2013, Khan et al.,
2014, Tamm et al., 2016, Awan et al., 2016 and Singh et al., 2016. Tyagi et al., 2014 used it for
sediment modelling and Pandey et al., 2014 and Tamm et al., 2016 used results of SWAT to
estimate hydropower potential of a catchment.

Many studies have attempted to simulate water quality variables by employing the SWAT
model. Jha et al., 2006 simulated streamflow, sediment losses, and nutrient loadings in the Raccoon
River watershed and assessed impacts of land use and management practice shifts. Hafiz et al.,
2012 examined applied the SWAT model to model flow, sediments and water quality parameters in
upper Thachin River Basin, Thailand with catchment area of 5,693 km$^2$. It was reported that the
model gave good results. Qiu and Wang, 2014 applied the SWAT model to the Neshanic River
watershed to simulate streamflow and water quality parameters including total suspended solids
(TSS), total nitrogen (TN), and total phosphorus (TP). An attractive feature of the SWAT model is
its ability to model the catchment response due to snow/glacier melt and rainfall. Many studies have
harnessed this feature of the model. For instance, Lemonds et al., 2007 calibrated the SWAT model
to the Blue River basin (867 km$^2$) in Colorado (USA) by adjusting the snowmelt, snow formation,
and groundwater parameters and obtained good fits to average monthly discharge values (NSE =
0.71). As per Stehr et al., 2009, the snow component of SWAT was capable of providing a
reasonably good description of the snow-cover extension over a small Chilean Basin (455 km$^2$).
Pradhanang et al., 2011 compared snow survey data for the catchment of Cannonsville reservoir
with model simulated snowpack and snowmelt at different elevation bands. When measured and
simulated snowpack were compared, correlation coefficients ranging from 0.35 to 0.85 were
obtained. Simulations of daily and seasonal streamflow improved when 3 elevation bands were
used. Troin and Caya, 2014 demonstrated the ability of SWAT to simulate snowmelt dominated



streamflow in the Outardes Basin, Quebec (Canada). The calibration of SWAT model showed a
satisfactory performance at the daily and seasonal time scales.

To improve the snow/glacier melt section of SWAT, some authors have attempted to
develop and plug-in routines for these processes. For example, Fontaine et al., 2002 developed a
snowfall-melt routine for mountainous terrain for the SWAT model which improved the correlation
between observed and simulated stream flow.Recently, Luo et al., 2013 proposed a dynamic
Hydrological Response Unit approach and incorporated an algorithm of glacier melt,
sublimation/evaporation, accumulation, mass balance and retreat into the SWAT model. They
simulated the transient glacier retreat and its impacts on streamflow at basin scale. This updated
model was applied in the Manas River Basin (MRB) in northwest China and the authors obtained
NSE of 0.65 for daily streamflow and small percent bias of 3.7% in water balance. The
hydrological community at large can make good use of such innovations if the relevant software
and guidelines for data preparation are made easily available.

Many other models or frameworks have been used in the recent past for modelling mountain
catchments. Shrestha et al. (2013) developed energy budget-based distributed modeling of snow
and glacier melt runoff in a multilayer scheme for different types of glaciers within a distributed
biosphere hydrological modeling framework. A study of Hunza River Basin (13,733 km$^2$) in the
Karakoram mountains where the SWAT model was used showed good agreement with observations
(NSE = 0.93). Likewise, Immerzeel et al. (2013) used results from an ensemble of climate models
along with a glacio-hydrological model for assessment of the impact of climate change on
hydrologic response of two Himalayan watersheds: the Baltoro (Indus) and Langtang (Ganges).
Future runoff was found to increase in both watersheds. A large uncertainty in future runoff arising
from variations in projected precipitation between climate models was noted. It is hoped that the
numerous attempts to apply the existing models to different geographies and research to develop
new modeling theories would lead to significant advances in hydrology of mountain basins across
the world.

Hydrological modeling of Himalayan river basins is important for many reasons. Nearly 2
billion people depend upon the waters of these rivers. Since most of these rivers are perennial and
the terrain has steep slopes, they have huge hydropower potential whose exploitation requires a
sound understanding of hydrologic response mechanism. Further, water triggered disasters are also
frequent in these basins. The region has complex topography and hydrologic data are scarce. In
addition, changes in land use/cover and climate are likely to significantly impact snow/glacier
accumulation and melt and hydrological response of these river basins. Snow and glacier melt
significantly contribute to flow of most Himalayan rivers and their modeling is an important
component in streamflow modeling of Himalayan rivers. Global warming is likely to accelerate
snow and glacier melt and it is necessary to study its impact for long term water resources planning.
However, in spite of well-recognized importance and need of such studies, not many attempts have
been made to assess hydrology of these rivers.



Clearly, there is a need for better understanding of the hydrologic response of the
Himalayan rivers for sustainable water management, developing the ability to forecast floods, and
predict the impacts due to changes in climate and land use/cover. To that end, a distributed model is
needed whose data requirements match with the availability. Although studies have been carried out
in mountainous catchments with a variety of topography, climate, and data availability by using the
SWAT model and the results have been quite good, only limited studies have been carried out in the
Indian Himalayan region. The hydrological and other data of this region that are needed for
modeling are not easily available and considerable efforts are required to collect and process the
data and setup a distributed model. Therefore, the objective of this study was to improve our
understanding of hydrological regime of the Himalayan rivers and enhance prediction of
hydrological processes. The main goal was achieved through carrying out hydrologic modeling of a
Himalayan river basin which receives contribution from snow/glacier and rainfall by employing
larger amount of observed data. We have also attempted to determine various water balance
components for better understanding of hydrologic response of the watershed. Better modeling and
hydrologic assessment of these basins will help in improved management of water resources,
harness hydropower potential, and partly overcome problems due to data scarcity. Such studies will
also help understand the likely impacts of climate change on water resources.
Before proceeding further the SWAT model is briefly described in the following.

## 2 THE SWAT MODEL

The Soil and Water Assessment Tool (SWAT) is a semi-distributed, continuous time watershed
modelling system which simulates hydrologic response of a catchment by using process-based
equations. It has been developed by the USDA Agricultural Research Service (Arnold et al., 1998).
Spatial variability in a catchment are represented in SWAT by dividing the catchment area into sub-
watersheds; these are further subdivided into hydrologic response units (HRUs). A HRU possesses
unique land use, soil types, slope and management practices (Neitsch et. al., 2002a, 2002b). To
computes the water balance, the model simulates a range of hydrologic processes such as
evapotranspiration, snow accumulation, snowmelt, infiltration and generation of surface and
subsurface flow components.
SWAT model allows division of maximum ten elevation zones in each sub-basin to consider
orographic effects on precipitation, temperature and solar radiation (Neitsch et al., 2001). Snow
accumulation, sublimation and melt are computed in each elevation zone and weighted average is
computed subbasin wise. Snowmelt depth in the same elevation band is assumed to be the same in
all sub-basins.

### 2.1 Modeling of Snowmelt

A temperature-index approach is used by SWAT model to estimate snow accumulation and
melt. Snowmelt is calculated as a linear function of the difference between the average snowpack
maximum temperature and threshold temperature for snowmelt. Snowmelt is combined with
rainfall while calculating infiltration and runoff. SWAT does not include an explicit module to
handle snow melt processes in the frozen soil, but includes a provision for adjusting infiltration and





estimating runoff when the soil is frozen (Neitsch et al., 2005). Despite this limitation, SWAT is
considered to be an appropriate integrated model for addressing a range of issues. It is noted that
many of the existing models do not have the capability to model both snow/glacier melt and
rainfall-runoff processes.
In the temperature-index approach, temperature is a major factor that controls snowmelt
(Hock, 2003). Snowmelt is computed as a linear function of the difference between average
snowpack maximum temperature and the threshold temperature for snowmelt, SMTMP:

$$SNO_{mlti} = b_{mlti} \cdot SNO_{covi} \left[ \frac{T_{snowi} + T_{maxi}}{2} - SMTMP \right]$$

(1)

Where $SNO_{mlti}$ is the amount of snowmelt on day i (mm $H_2O$), $T_{maxi}$ is the maximum air
temperature on day i (°C), SMTMP (°C) is snowmelt base temperature above which snow will be
allowed to melt and $b_{mlti}$ is the melt factor on day i (mm $H_2O$-day). Snowmelt is included with
rainfall in computation of infiltration and runoff.
The classification of precipitation is based on a threshold value of mean air temperature. If
the average daily air temperature is below the snowfall temperature, the precipitation in a HRU is
considered as solid (or snow) and the liquid water equivalent of the snowfall is added to snowpack.
The snowpack is depleted by snowmelt or sublimation. The mass balance for the snowpack for a
HRU is:

$$SNO_i = SNO_{i-1} + P_s - E_{subi} - SNO_{mlti}$$

(2)

where, $SNO_i$ is the water content of the snowpack (mm $H_2O$), $P_s$ is the water equivalent of snow
precipitation (mm $H_2O$), $E_{subi}$ is the amount of snow sublimation (mm $H_2O$), and $SNO_{mlti}$ is the
water equivalent of snow melt (mm $H_2O$), all for day $i$.
The spatial non-uniformity of the areal snow coverage over the HRU is taken account
through an areal snow depletion curve that describes the seasonal growth and recession of the
snowpack (Anderson, 1976). Two addition parameters are defined at the watershed scale,
SNOCOVMX and SNO50COV. These control the areal depletion curve by accounting for the
variable snow coverage as:

$$SNO_{covi} = \frac{SNO_i}{SNOCOVMX} \left[ \frac{SNO_i}{SNOCOVMX} + \exp\left( cov_1 - cov_2 \cdot \frac{SNO_i}{SNOCOVMX} \right) \right]^{-1}$$

(3)

where $SNO_{covi}$ is the fraction of HRU area covered by snow on the day $i$, $SNO_i$ is the water content
of the snow pack on day $i$, SNOCOVMX is the minimum snow water content that correspond to
100% snow cover (mm $H_2O$), and $cov_1$ and $cov_2$ are coefficients that control the shape of the curve.
**2.2 Modeling of Catchment Hydrology**
Weather, soil properties, topography, vegetation and land management practices are the most
important inputs for the SWAT model. SWAT computes actual soil water evaporation using an
exponential function of soil depth and water content. The modified Soil Conservation Service





(SCS) curve number method is used to compute runoff. The influence of plant canopy infiltration
and snow cover is incorporated into the runoff calculation. To support soil water processes such as
infiltration, evaporation, plant uptake, lateral flow, and percolation to lower layers, the soil profile
is subdivided into many layers. When field capacity of a soil layer is exceeded downward flow
occurs and the layer below is not saturated. Percolation from the bottom of the soil profile recharges
the shallow aquifer. Lateral sub-surface flow in the soil profile is calculated simultaneously with
percolation. Groundwater flow contribution to total stream flow is simulated by routing the shallow
aquifer storage component to the stream. Runoff is routed through the channel network by the
variable storage routing method or the Muskingum method (Neitsch et al., 2005).
SWAT model simulates hydrologic cycle based on the water balance equation:

$$SW_t = SW_o + \sum_{i=1}^{n} (R_{day} - Q_{surf} - E_a - w_{seep} - Q_{gw})$$

(4)

where, $SW_t$ is the final soil water content (mm $H_2O$), $SW_o$ is the initial soil water content (mm
$H_2O$), t is time in days, $R_{day}$ is amount of precipitation on day i (mm $H_2O$), $E_a$ is the amount of
evapotranspiration on day i (mm $H_2O$), $Q_{surf}$ is the amount of surface runoff on day i (mm $H_2O$),
$w_{seep}$ is the amount of percolation and bypass exiting the soil profile bottom on day i (mm $H_2O$),
and $Q_{gw}$ is the amount of return flow on day i (mm $H_2O$).
Since the model maintains a continuous water balance, the subdivision of the watershed in
HRUs enables the model to consider differences in evapotranspiration for different crops and soils.
Runoff is predicted separately for each sub area and is routed to compute total runoff for the basin.
SWAT model software and documentation are freely available through Internet at
http://swat.tamu.edu/software/swat-executables/.

**2.3 Temperature index with elevation band approach**
This method incorporates elevation and temperature which is used to determine the snow
pack and snowmelt caused by orographic variation in precipitation and temperature. Many studies,
e.g., Zhang et al. (2008), have shown that elevation is an important factor in the variation of
temperature and precipitation. Fontaine et al. (2002) introduced a modified snowfall-snowmelt
routine for mountainous terrain into SWAT. This modified routine allows the SWAT model to
divide each sub-basin into 10 elevation bands and simulates the spatial and temporal variation of
snowpack and snowmelt on account of elevation. The temperature and precipitation for each
elevation band was adjusted by using:

$T_B = T + (Z_B - Z) \cdot dT / dZ$  (5)
$P_B = P + (Z_B - Z) \cdot dP / dZ$  (6)





Where, $T_B$ is the mean temperature (ºC) in the elevation band, T is the temperature measured at the
weather station (ºC), $Z_B$ is the midpoint elevation of the band (m), Z is the elevation (m) of the
weather station, P is the precipitation measured at the weather station (mm), $P_B$ is the mean
precipitation of the band (mm), dP/dZ is the precipitation lapse rate (mm∕km), and dT/dZ is the
temperature lapse rate (ºC∕km).

## 3   THE STUDY AREA AND DATA USED

In the present study, Ganga River Basin up to Devprayag have been considered. The study area lies
in the North- Western Himalayan ranges, between latitudes 30º to 31º 30' North and longitudes 78º
7' to 80º 15' East in India and is shown in Figure 1.The size of the catchment is about 18728
km²and elevation varies from 427 m to 7785 m. Bhagirathi and Alaknanda Rivers are the two
headwater streams that join at Devprayag to form Ganga River. The Bhagirathi River originates
from the snout of the Gangotri Glacier at Gomukh (3900 m). It flows for 217 km to reach
Devprayag and is joined by Bhilangana and Asiganga Rivers on the way. Asiganga joins Bhagirathi
River at 5 km upstream (1120 m) of Uttarkashi from west direction. Bhilangana River originates
from Khatling glacier (3950 m) and joins the Bhagirathi River at Tehri from east direction.
Alaknanda River rises at the confluence and the foot of the Satopanth and Bhagirath Kharak
Glaciers. The Alaknanda River flows for about 224 km before meeting with Bhagirathi River at
Devprayag. Its main tributaries are Dhauli Ganga, Pindar, Nandakini and Mandakini. The average
rainfall in the study area varies between 1000 to 2500 mm, of which 60-80% falls during the
monsoon period between June and September. The rivers experiences strong seasonal climatic
variations, which is also reflected in the monthly variation in stream flows. High flow takes place
during June-September, when the combined influence of rainfall and snow melt is at the maximum.
In this study, a number of maps have been prepared. The sources and resolution of ASTER
DEM, land use land cover map and soil map are given in Table 1.
Meteorological data for the study area consisted of 16 years of time series (1990-2005) of
daily precipitation, minimum and maximum temperature, solar radiation and wind speed for 7
stations, namely Badrinath, Joshimath, Karanprayag, Rudraprayag, Uttarkashi, Tehri and
Devprayag. The rainfall data obtained for these stations were having many gaps. Therefore,
precipitation data from *Asian Precipitation - Highly-Resolved Observational Data Integration*
*Towards Evaluation of Water Resources* (APHRODITE's Water Resources) were used. The
APHRODITE project develops state-of-the-art daily precipitation datasets with high-resolution
grids for Asia. The datasets are created primarily with data obtained from a rain-gauge-observation
network. APHRODITE's Water Resources project has been conducted by the Research Institute for
Humanity and Nature (RIHN) and the Meteorological Research Institute of Japan Meteorological
Agency (MRI/JMA) since 2006. A daily gridded precipitation dataset for 1961-2007 was created by
collecting rain gauge observation data across Asia through the activities of the APHRODITE
project (http://www.chikyu.ac.jp/precip/).The final data product does not have any gaps.
APHRODITE data is available in the form of grid of 0.5°×0.5° and 0.25°×0.25°. The data of
0.25°×0.25° was downloaded for the period of 1961-2007 and converted into map form using





ArcGIS and exported to ERDAS Imagine. The grids for which data were downloaded and used in
this study include Devprayag (altitude 469m), Tehri (608m), Rudraprayag (612m), Karnprayag
(784m), Joshimath (1446m) and Badrinath (3136m).

Daily stream flow data collected from Central Water Commission (CWC) for gauging
station located in the study area were used for model calibration and validation purpose. The
discharge gauging station Devprayag-Z9 is located downstream of the confluence of Bhagirathi and
Alaknanda rivers at Devprayag. The data measured at Devprayag-Z9 site were used to model the
Ganga River Basin up to Devprayag.

**3.1 Land Use and Soil Data**
Land use is one of the most important factors affecting runoff, soil erosion and evapotranspiration
in a watershed (Neitsch et al., 2005).
In the Ganga basin up to Devprayag, the most dominant land use/ land cover are open forest,
dense forest, barren land, snow cover area and range land covering 23.28%, 20.99%, 13.41%,
36.4% and 5.9% of the total basin area, respectively (Figure 2). For soils, Orthents (66%),
typicudorthents (21%) and typiccryochrepts (12%) are the most dominant soils having 2, 3 and 3
layers respectively in the basin (Figure 3).

**3.2 Model Set Up**
In the setup of SWAT model for the study catchment, the first step is identification and delineation
of hydrological response units (HRUs). River network for Ganga basin up to Devprayag were
delineated from ASTER DEM by using the analytic technique of the ArcSWAT 2009 GIS
interface(Figure 4).To obtain a reasonable numbers of HRUs within each subbasin, a unique
combination of landuse and soil (thresholds of 10% in land use/land cover and 5% in soil type)
were used. In this procedure, the Ganga River Basin was divided into 7 sub-basins and 126 HRUs
as shown in Figure 5. These set up ensures a stream network definition that satisfactorily represents
the dominant land uses and soils within each subbasin and at the same time, a reasonable number of
HRUs are created in each sub-basin.

The SWAT model has a large number of parameters that describe the different hydrological
taking place in the study basin. During calibration process, model parameters were systematically
adjusted to obtain results that best match with the observed values. In the validation process, the
catchment response was simulated by using the parameters finally obtained during the calibration
process. For evaluating the model performance computed hydrographs was compared with the
observed hydrograph. It may be stated here that the streamflow data for the Ganga basin is
classified and cannot be disclosed. Hence, we have shown scaled values of the flows in various
graphs of Ganga basin.

The length of calibration data is an important factor in model calibration. The available data
is usually partitioned in two sets: calibration data and validation data. Usually, calibration is carried
out by using more years of data; say about 60 – 75 % of the available data. Typical questions that
arise in this respect are: how much data are necessary/enough to obtain a good model calibration




and what are the characteristics that the calibration data should have to maximize the chances of obtaining reliable parameter estimates? Ideally, model calibration should result in parameter values that produce the best overall agreement between simulated and observed values (discharge in this case). Yapo et al., 1996 found that for the watersheds similar to their study area, approximately 8 years of data may be necessary to obtain a calibration that is relatively insensitive to the period selected and that the benefits of using more than 8 years of calibration data may be marginal. Regarding the characteristics, parameter identifiability significantly improves when the all the hydrologic components are activated during the calibration period.

Statistical performance measures of the hydrological models are computed to determine how the values simulated by the model match with those observed. For this study, the statistical criteria that were used to evaluate model performance were the goodness-of-fit ($R^2$), the Nash-Sutcliffe efficiency index (NSE) and coefficient of regression line multiplied by the coefficient of determination ($bR^2$). The model performance is considered to be better as the values of $R^2$ and NSE approach unity.

The observed daily stream flow data from year 1990 to 1998 were used to calibrate the SWAT model and the model was validated by using the data from the year 1999 to 2005. Data for the first two years (1990 and 1991) were reserved as "warm-up" period (to overcome the errors due to incorrect initial conditions, the results of model run for a few initial periods are not used in analysis of results. These initial periods are termed as the warm up period). Thus the model calibration statistics was evaluated for the period 1992-1998.

## 4 RESULTS AND DISCUSSIONS

Initially, the SWAT-CUP which uses Sequential Uncertainty Fitting (SUFI2) algorithm developed by Abbaspour et al., 2007 was used in this study. SUFI2 is a multi-site, semi-automated global search procedure for model calibration and uncertainty analysis. The sources of uncertainties which includes temperature and rainfall parameters and measured data are accounted for in SUFI2. SUFI2 uses P-factor, the percentage of measured data bracketed by the 95% prediction uncertainty (95PPU), and the R-factor average width of the 95PPU band divided by the standard deviation of the measured data, to assess uncertainty. Abbaspour et al., 2007 have described SUFI2 algorithm in detail.

While studying the SUFI2 calibration results, it was seen that in the table of monthly values of various water balance components produced by SWAT, snowfall had fairly high values in monsoon months whereas the study area does not receive snowfall in monsoon months (June to September). PLAPS and TLAPS were the parameters controlling the temporal distribution of precipitation (whether rain or snow). The calibrated values of PLAPS and TLAPS from the SWAT-CUP were 8.5 and -5.83. To have realistic values of precipitation distribution, TLAPS and PLAPS were changed systematically and TLAPS equal to -4.0 and PLAPS equal to 8.55 yielded precipitation values which were realistic for the study area. However, as a result of this change,





there were many significant deviations between observed and simulated hydrographs. In particular, the recession limbs and hydrograph during lean season had poor match as shown in Figure. 6.

At this stage, sensitivity analysis of model parameters was performed. Twenty calibrated parameters including seven snowmelt related parameters were used for sensitivity analysis. The sensitivity rank, default value, range of the parameter values and the optimal values for Devprayag sites in the basin are given in Table 2. The remaining parameters were not much sensitive for the model output.A number of SWAT parameters are related to snow but out of these, three parameters, viz. maximum temperature index melt factor SMFMX, the snowmelt base temperature SMTMP and the minimum temperature index melt factor SMFMN were found to be important.

The parameters were ranked in terms of their sensitivity to the model calibration. These sensitive parameters were mainly responsible for changes in model output during calibration. Results showed that CN is the most sensitive parameter to changes in discharge. Next to CN, the other parameters that were found to be sensitive are those related to soil and groundwater. Among these, ALPFA_BF is the baseflow recession constant.

For proper simulation of the hydrograph during post monsoon and lean season, the changes in hydrograph in response to changes in several key parameters were also studied. It is noted during sensitivity analysis that changes in some model parameters may cause different type of changes in the simulated discharge in different seasons (monsoon and post monsoon). For instance, if the value of CN is increased, the simulated streamflow increased from September to May but decreased from June to August. When the value of ALPHA_BF was reduced, the simulated streamflow was found to increase from October to March and decrease from April to September. When the value of SOL_AWC was decreased, the simulated streamflow increased from September to February and decreased from March to August. When the value of SOL_K was increased, the simulated streamflow increased from February to August but decreased from September to January. When the value of GW_DELAY was increased, the simulated streamflow increased from March to August and decreased from September to February.

To improve the match between observed and simulated hydrographs in terms of recession limb and base flow, it was hypothesized that the movement of water through soil and ground water zone is not being properly modeled. Since baseflow was being under-simulated, more water should be allowed to enter the sub-surface zone, stay there for some time, and then emerge as baseflow. Accordingly, several simulations were carried out by changing the soil and ground water related parameters till the simulated hydrographs shows a good match with the observed hydrograph. The parameters that were systematically tuned include ALPHA_BF, SOL_AWC, SOL_K, and GW_DELAY. Two statistical performance measures (coefficient of determination $R^2$ and NSE) and visual inspection of the plot between observed and computed hydrographs were used to evaluate the performance of the model in simulating streamflows and to decide which parameters to change.

The hydrographs of the observed and simulated daily and monthly flows for the calibration period (1992-1998) for the Ganga Basin up to Devprayag are shown in Figure 7a and 9a. It is seen




that the overall shape of the simulated hydrograph is matching well with the observed hydrograph
and the recession behaviour is also well simulated now. A few of the observed high peaks have
been simulated well while some of the peak values do not match well. The time series of the
observed and simulated daily and monthly hydrographs for the validation period are shown in
Figure 8a and 10a. It is seen from the graph that the simulated hydrograph correlates significantly
well with observed hydrograph. Scatter plot between observed and simulated daily and monthly
discharges for the calibration data (Figure 7b and 9b) indicate even distribution of most of the
points around the 1:1 line. Of course, a few data points are away from the line. Further, for the
validation period, the scatter plot for daily and monthly (Figure 8b and 10b) also shows the points
of the simulated flows are close to the 45$^o$ line.
The statistical performance indicators for calibration and validation for the Ganga basin up
to Devprayag are given in Table 3. The coefficient of determination ($R^2$) was 0.69 and 0.95 for
daily and monthly calibration period and 0.57and 0.94 for daily and monthly during validation
period. The NSE was computed as 0.64 and 0.80 for daily and monthly calibration period and
0.49and 0.85for daily and monthly validation period. Thus one may conclude that the indices for
monthly data are excellent and can be termed as very good for the daily data. Figure 11 show the
plot of observed and computed hydrograph for one year. It can be seen that some peaks have been
properly simulated but some are not. It is highlighted that the density of raingauge network in the
study area is grossly inadequate. Due to this many rainfall events that occur in the vicinity
contribute large volumes of water in the model even though their spatial coverage may be small. On
the other hand, if the event does not occur around the raingauge will be missed even though it may
have a large spatial coverage.
For calibration and validation, various water balance components are given in Table 4. The
water balance components include: the total amount of precipitation, actual evapotranspiration,
snowmelt runoff, and water yield. Here, water yield includes surface runoff, lateral flow to stream
and water from shallow aquifer that returns to river reach. The results indicate that contribution
from direct surface runoff is small in the water yield and the main contribution to water yield is
through lateral flow and ground water flow. ET comes out to be 43-46% of precipitation. As
catchment of Ganga river up to Devprayag site has comparatively less snow covered area, ET is at
higher rate. The snowmelt runoff contribution at Devprayag site comes out to be 20% and 13% of
the water yield during calibration and validation respectively. The water yield, i.e. sum of surface
runoff, lateral runoff and ground water contribution in stream flow comes out be about 57-58%of
the precipitation. In the basin, interflow contributes significantly to the water yield as compared to
shallow groundwater.

## 5  CONCLUSIONS

This study has attempted to simulate the response of hilly parts of a Himalayan river basin, viz., the
Ganga basin up to Devprayag. The values of R$^2$ and NSE for calibration (1992-1998) and validation
(1999-2005) vary between 0.69and 0.64 and can be considered as good (as per Moriasi et al., 2007)



for the basin given the availability of meteorological and pedological data. Overall, the hydrograph
shape could be reproduced satisfactorily although all the peaks and the recession limbs could not be
reproduced very well. Thus, the SWAT model can be considered to be a good tool to model the
discharge hydrograph and various water balance components for a Himalayan basin.

Water yield for the basin is ranging between 57-58% of the precipitation. Snow/glacier melt
contribution is 13-20% for the Ganga basin. In the Ganga basin, interflow contributes significantly
to the water yield. However, these results are required to be buttressed by more detailed hydrologic
modeling of some more river basins to investigate their response mechanism. To that end, more
attention is needed to strengthen spatial, soil and hydrometeorological database including snowfall
for the study basins by installing automatic weather stations to measure precipitation (rain and
snow) and other climatic variables at various elevations. Isotope analysis may be carried out to
separate the runoff components and compare the results with hydrologic model.

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





Table 1: Sources and description of the input data for the Ganga Basin up to Devprayag

| Data Type | Source | Spatial/ Temporal Resolution | Description |
| --- | --- | --- | --- |
| Topography | http://gdem.ersdac.jspacesystems.or.jp | 90 m | Aster Digital Elevation Model |
| Land use/Land cover | www.iitd.ac.in | 90 m | Land-use Classification |
| Soils | www.iitd.ac.in | 90 m | Soil Classification |
| Weather Data | APHRODITE DATA | 1.0 deg 0.25 deg | Temperature, Solar Radiation, Wind Speed, Precipitation, Data |
| Stream Flow | Central Water Commission, Dehradun | Daily | Daily stream flows measured at the gauging stations |

Table 2: Description of 20 calibration parameters: Sensitivity rank, Default value, Range and
Optimal value

| Parameters | Description | Sensitivity Rank | Default value | Range | Optimal value |
| --- | --- | --- | --- | --- | --- |
| CN2 | Initial SCS runoff curve number for moisture condition II | 1 | 77-92 | 35-98 | 55-92 |
| TLAPS | Temperature Lapse Rate | 2 | 0 | -10 to 10 | -4.0 |
| SOL_AWC | Available water capacity of the soil layer (mm/mm) | 3 | 0.067 – 0.146 | 0 - 1 | 0.024 – 0.053 |
| SOL_K | Saturated hydraulic conductivity of soil (mm/hr) | 4 | 1.95 – 121.12 | 0 - 2000 | 3.0 – 186.61 |





| | | | | | |
|---|---|---|---|---|---|
| SOL_Z | Soil depth (mm) | 5 | 60 - 170 | 0 -3500 | 30 - 300 |
| ALPHA_BF | Base flow alpha factor (days) | 6 | 0.084 | 0-1 | 0.07 |
| GW_DELAY | Groundwater delay (days) | 7 | 31 | 0-500 | 35 |
| REVAPMN | Threshold depth of water in the shallow aquifer for "revap" to occur (mm) | 8 | 1 | 0-500 | 499 |
| RCHRG_DP | Deep aquifer percolation fraction | 9 | 0.05 | 0-1 | 0.015 |
| GW_REVAP | Groundwater revap coefficeient | 10 | 0.02 | 0.02 - 0.20 | 0.162 |
| CH_K2 | Effective hydraulic conductivity in main channel alluvium (mm/h) | 11 | 0 | -0.01 - 500 | 20.43 |
| ESCO | Soil evaporation compensation factor | 12 | 0.95 | 0-1 | 0.40 |
| PLAPS | Precipitation Lapse Rate | 13 | 0 | -1000 to 1000 | 8.55 |
| SMTMP | Snowmelt base Temperature ($^{o}$C) | 14 | 0.5 | -5 to 5 | -3.05 |
| SMFMN | Minimum melt rate for snow during year (mm $H_2O$/ $^{o}$C -day) | 15 | 4.5 | 0 - 10 | 0.80 |
| SMFMX | Maximum melt rate for snow during year (mm $H_2O$/ $^{o}$C -day) | 16 | 4.5 | 0 - 10 | 4.79 |
| SNO50COV | Snow water content corresponding to 50% snow cover | 17 | 0.50 | 0 - 1 | 0.48 |
| SFTMP | Snowfall Temperature ($^{o}$C) | 18 | 1 | -5 to 5 | -2.79 |
| TIMP | Snow pack temperature lag factor | 19 | 1 | 0-1 | 0.394 |
| SNOCOVMX | Minimum snow water content corresponding to 100% snow cover, SNO100 (SNOCOVMX- mm $H_2O$) | 20 | 1 | 0 - 500 | 242.40 |






Table 3: Statistical performance indicators for calibration and validation for Ganga River Basin up
to Devprayag

| Daily/Monthly | Calibration (1992-1998), Validation(1999-2005) | $R^2$ | NSE |
|---|---|---|---|
| Daily | Calibration | 0.69 | 0.64 |
| | Validation | 0.57 | 0.49 |
| Monthly | Calibration | 0.95 | 0.80 |
| | Validation | 0.94 | 0.85 |

Table 4: Water balance components in mm

| | Precipitation | ET | Surface Runoff | Lateral flow | Ground water flow | Water yield | Snow fall | Snow melt |
|---|---|---|---|---|---|---|---|---|
| Calibration | 1236.1 | 484.8 | 96.63 | 297.30 | 293.92 | 686.87 | 140.74 | 73.84 |
| Validation | 1203.3 | 512.1 | 74.24 | 299.37 | 290.05 | 662.81 | 110.57 | 48.87 |







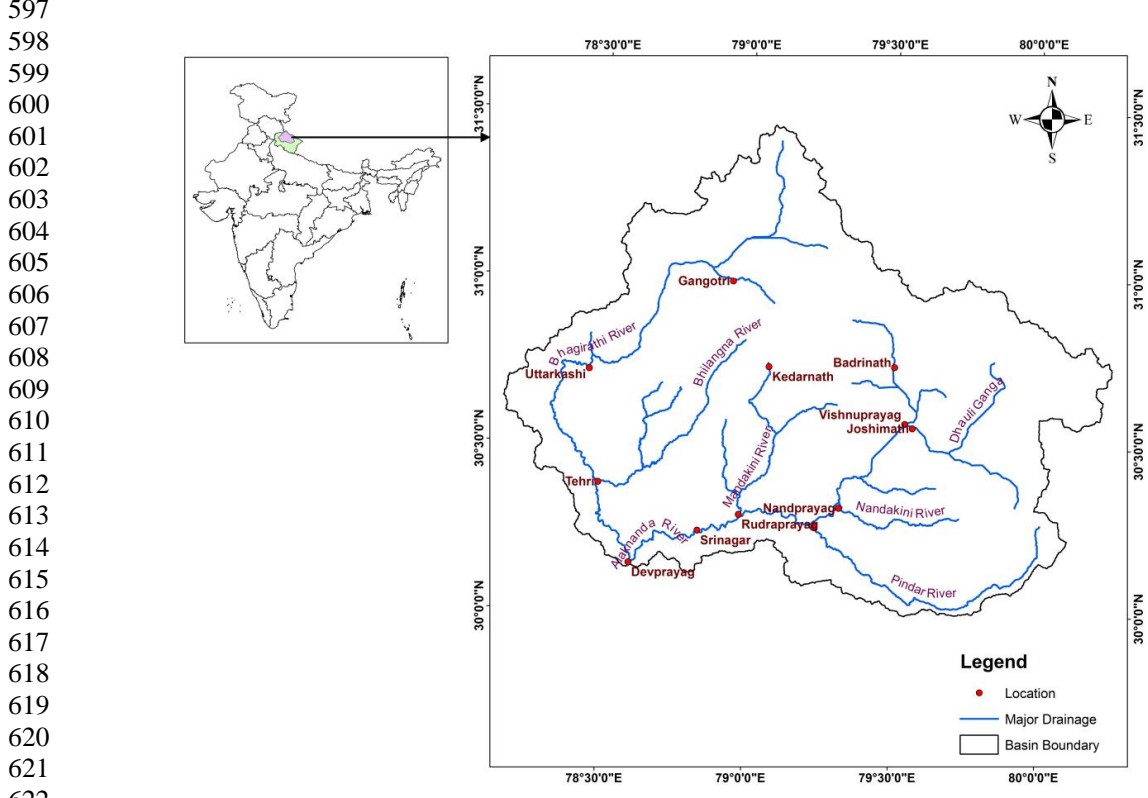

Figure 1: Index map of Ganga basin up to Devprayag


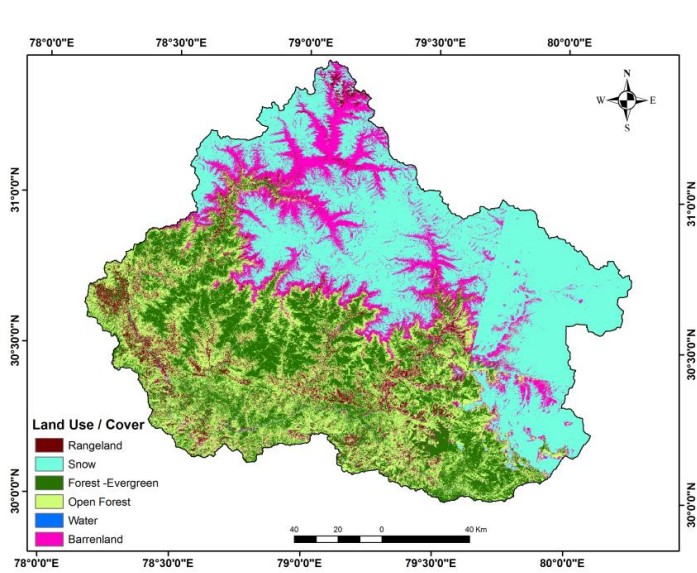

Figure 2: Land use/cover of Ganga basin up to Devprayag

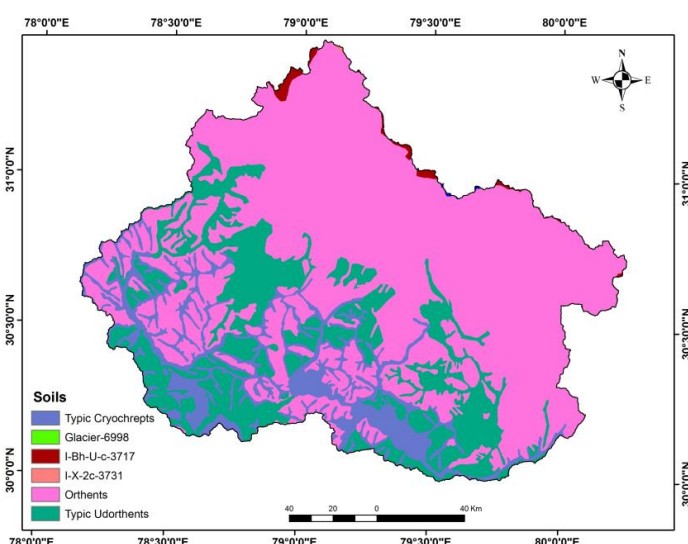

Figure 3: Soil map of Ganga basin up to Devprayag



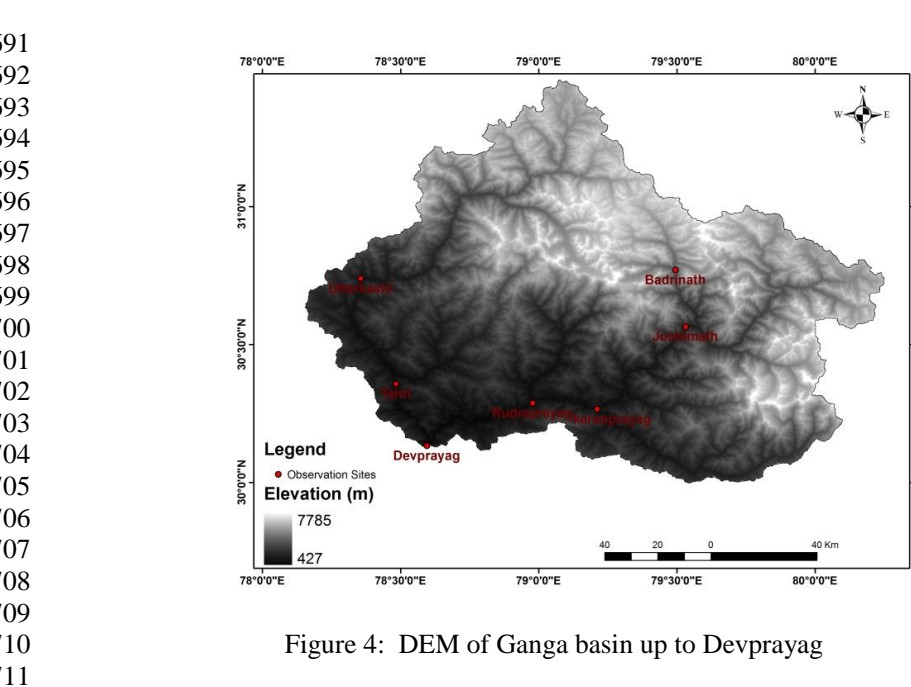

Figure 4:  DEM of Ganga basin up to Devprayag

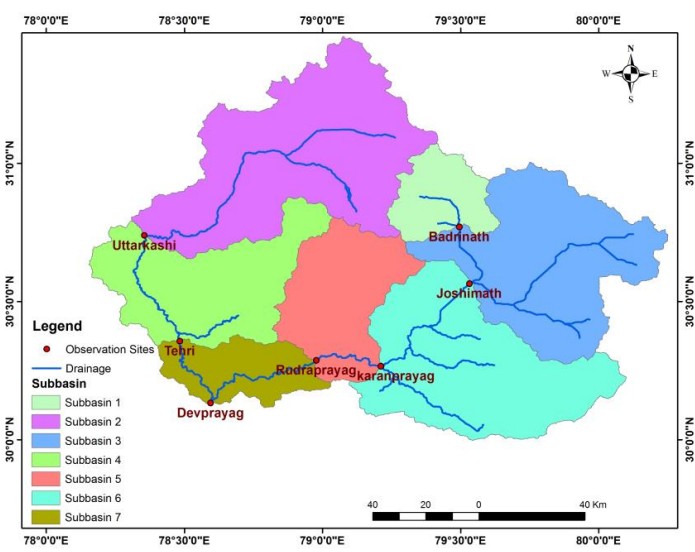

Figure 5: Sub basin map of Ganga basin up to Devprayag





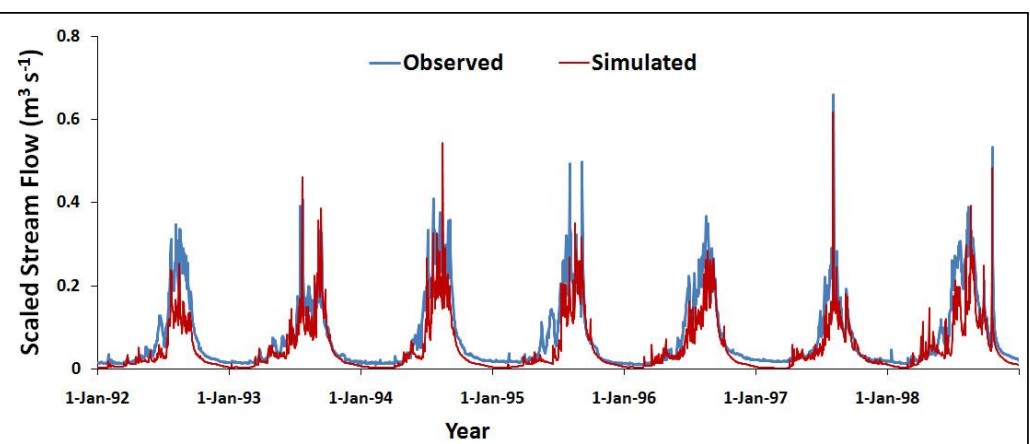

Figure 6: Comparison of daily observed and simulated stream flow hydrograph of Ganga basin up to Devprayag during calibration period (1992-1998) during iteration process






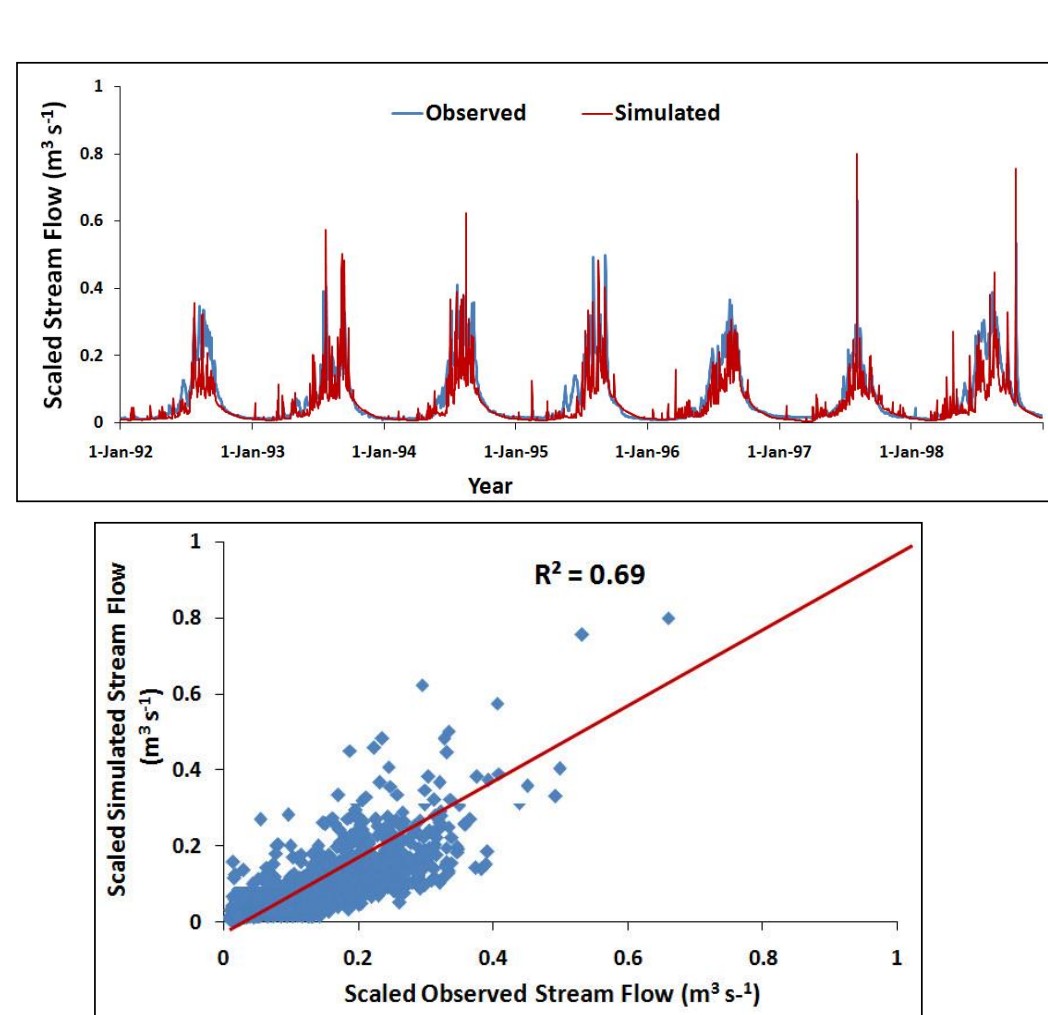

Figure 7: Comparison of a) daily observed and simulated stream flow hydrograph of Ganga basin
up to Devprayag during calibration period (1992-1998), and b) scatter plot




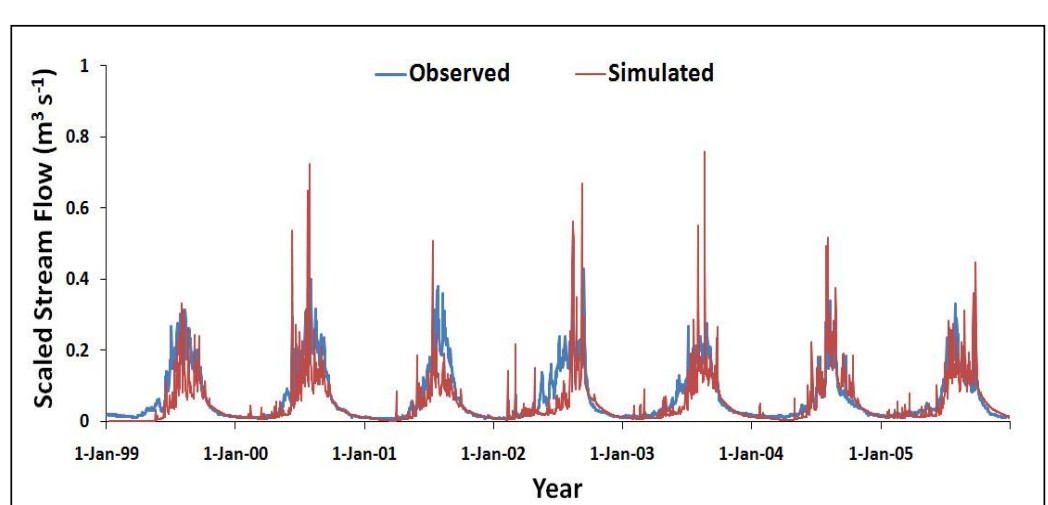

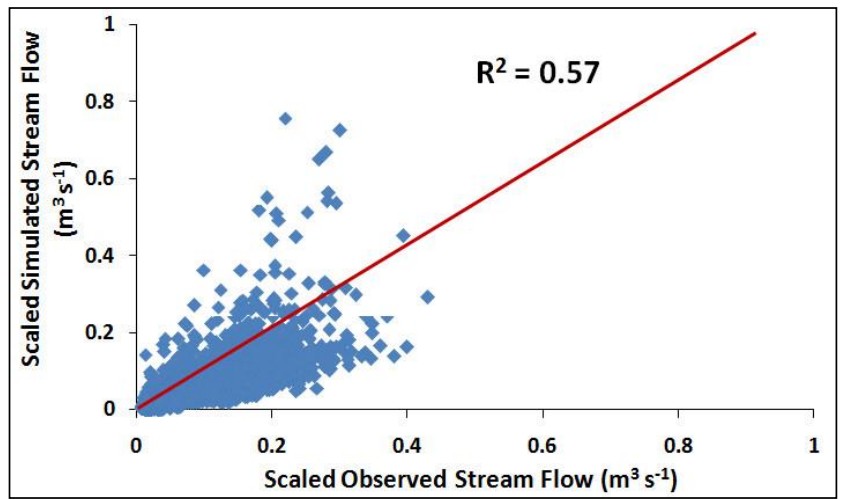

Figure 8: Comparison of a) daily observed and simulated stream flow hydrograph of Ganga basin
up to Devprayag during validation period (1999-2005), and b) scatter plot





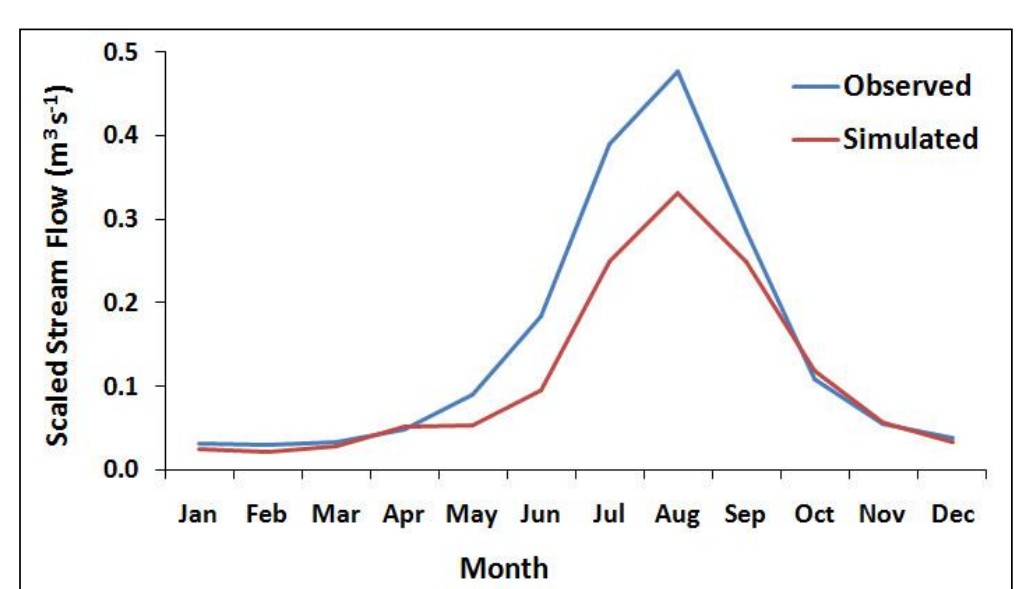

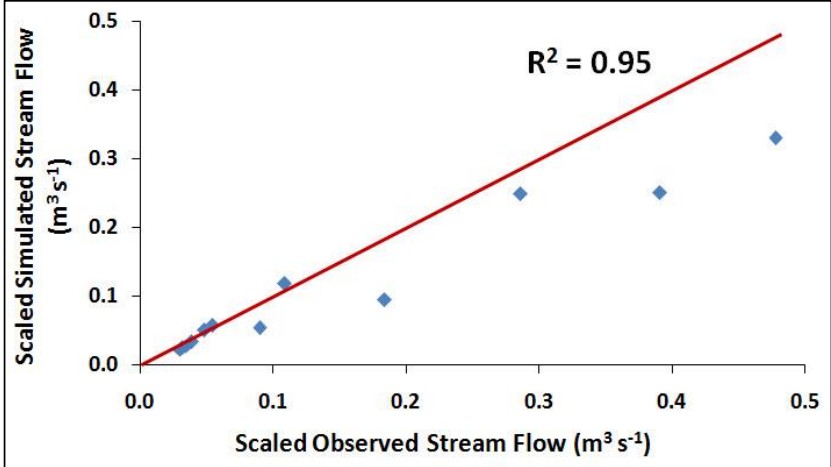

Figure 9: Comparison of a) Monthly observed and simulated stream flow hydrograph of Ganga basin up to Devprayag during calibration period (1992-1998), and b) scatter plot






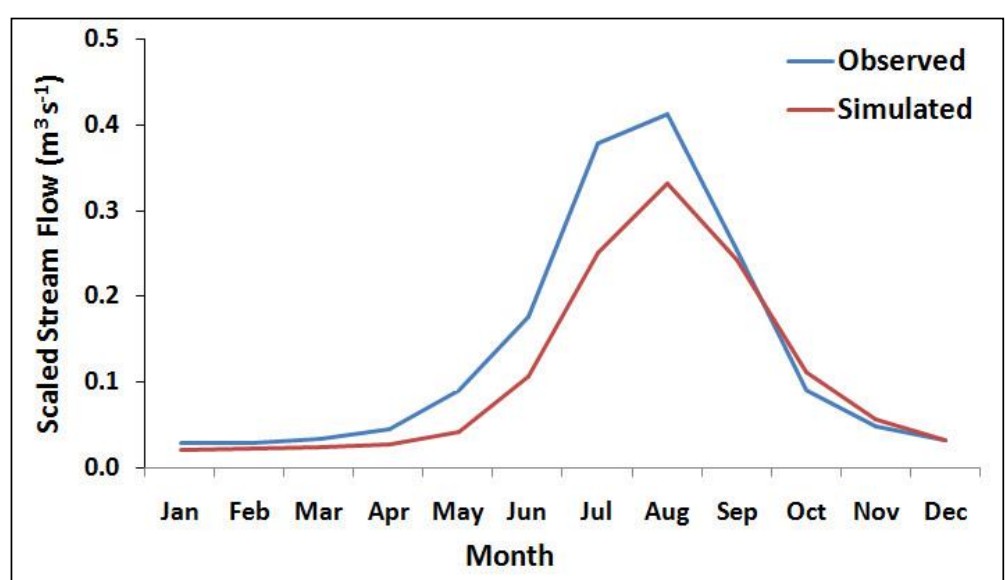


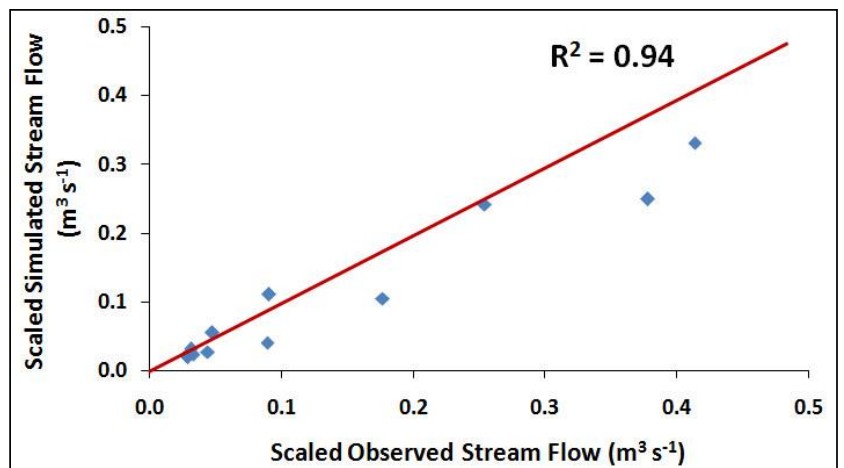

Figure 10: Comparison of a) Monthly observed and simulated stream flow hydrograph of Ganga
basin up to Devprayag during Validation period (1999-2005), and b) scatter plot.






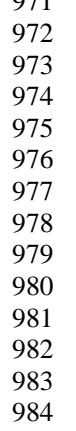
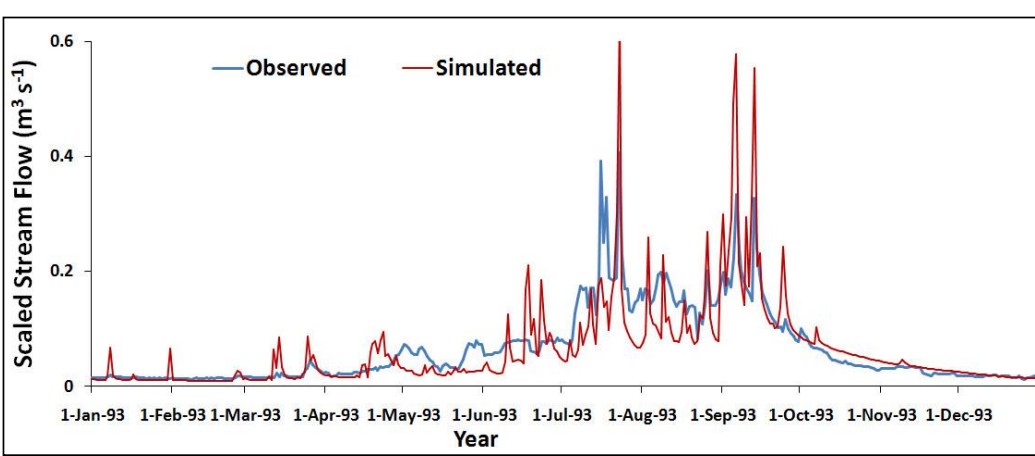

Figure 11: Daily observed and simulated stream flow hydrograph of Ganga basin up to Devprayag
for the year 1993.