# Peer review of "Hydrologic modeling of a Himalayan mountain basin by using the SWAT model Sharad K. Jain1\*, Sanjay K. Jain1, Neha Jain1 and Chong-Yu Xu2 1National Institute of Hydrology, Roorkee 247667, India 2 University of Oslo, Oslo, Norway \*Corresponding Author: Email s\_k\_jain@yahoo.com"

_Hydrology and Earth System Sciences, 2017_

## Referee Comment (RC1) · M. Masood (Referee) · 16 Apr 2017

Although some of my comments below are critical, I should acknowledge that the authors have put enormous effort in undertaking this study and I congratulate them for the work. The manuscript has been focused in details on hydrologic modeling of a part of Ganga basin which is a Himalayan mountain basin. Authors chose SWAT model to model the basin to achieve the aim of understanding hydrological responses of the basin. I believe this paper is also relevant to the special issue "The changing water cycle of the Indo-Gangetic Plain". However, I would like to recommend the authors to revise the manuscript thoroughly by addressing following issues.

General comments on Text

1. I found the novelty of this study is very limited to publish it in a high impacted journal

like HESS. There are many study already conducted on mountain basin in different parts of the world using various hydrologic model including SWAT. Several those previous studies are also discussed by the authors in the Introduction. However, it is difficult to find the uniqueness of their study among those studies. Therefore, I think, authors should identify the novelty of the study and should highlight it in Introduction. 2. In the Introduction, the authors presented literature review in great detail. However, I think, some studies should not be mentioned here as these are not relevant with current study. For instance, the paragraph in page-3, line 82-92 can be removed. 3. As I understand the central aim of the study is to improve the understanding of hydrological processes of the mountain basin. To achieve that goal, the authors have just calibrated and validated the SWAT model on the basin. I think this is not enough to understand the whole hydrological processes. The study needs in depth analyses of all hydrologic and climatic components and relationship among the components. They should justify/compare their results with existing previous studies. 4. If their objective is just hydrologic modeling of the basin, then it is better to incorporate the following additional analysis to improve the paper. (a) update/modify any module of the SWAT model and then apply it or (b) include addition analysis on model parameter uncertainty, sensitivity. The following paper may help Masood, M, Yeh, P. J. F., Hanasaki, N., and Takeuchi, K.: Model study of the impacts of future climate change on the hydrology of Ganges–Brahmaputra–Meghna basin, Hydrology and Earth System Sciences, 19(2), 747-770, doi:10.5194/hess-19-747-2015, 2015c.2. 5. NSE of daily simulated hydrograph for calibration and validation is 0.57 and 0.49, respectively, which are below satisfactory. The authors should conduct additional parameter sensitivity analysis to find better parameter values aiming better model performance. 6. Throughout the manuscript, this group of words "Ganga basin up to Devprayag.." has been repeated several times. Please avoid this repetition. 7. The model was simulated using bit old data (1992-2005). Why don't they choose recent data?

General comments on Figures

1. Overall quality of figures should be improved. 2. Sub-title with figure number (a) and (b) should be placed in all sub-plots of all figures with multi-plots. 3. Statistical indices (NSE, coefficient of determinant.. etc.) should be put in the relevant figures. 4. Importance of presenting the scatter plots (Fig. 7b. 8b) is limited; because well defined trend is not observed in those figures.

Specific comments on Text

1. Page-4, line-120: predict => project. 2. Page-4, line-121: Although studies... => Although many studies... 3. Page-7, line-236: Please provide a space in between km2 and. 4. Page-8, line-295: model parameters were systematically adjusted. Please discuss in details which procedure were followed for model parameter adjustment. 5. Page-8, line-299: It is mentioned that the streamflow data of the study basin is classified and cannot be disclosed. Please provide the reference of this statement. 6. Page-8, line-300: Please discuss how the streamflow were scaled down. 7. Page-9, line-329: Please provide a space in between study. SUFI2. 8. Page-9, line-336: Please provide a space in between the SUFI2. 9. Page-10, line-345: Please remove dot after Figure. 10. Page-10, line-350: Please provide a space in between output. A. 11. Page-10, line-378: visual inspection is not a good procedure to judge the performance. Please calculate more other statistical indices such as correlation coefficient, root-mean square error (RMSE). 12. Page-10, line-381: 9a. => 9a, respectively. 13. Page-11, line-386: 10a. => 10a, respectively. 14. Page-11, line-394: Please provide a space in between 0.57 and. 15. Page-11, line-396: Please provide a space in between 0.49 and. 16. Page-11, line-397: show => shows. 17. Page-11, line-400: this => these. 18. Page-11, line-420: Please provide a space in between 0.69 and.

Specific comments on Figures

1. Fig. 4: As the elevation range is very high, multi-color gradient can be chosen instead of current two color (black and white) gradient. 2. Fig. 6 should be removed; because same figure is put in the Fig. 7(a).

Specific comments on Tables

1. Table 1: References of the data sources are not appropriate. Please provide the references of relevant publications/report instead of web link. www.iitd.ac.in is an university web address. How can it be a data source? Please provide the specific data reference. For instance, the reference of APHRODITE data is Yatagai et al., 2012.

Yatagai, A., Kamiguchi, K., Arakawa, O., Hamada, A., Yasutomi, N., and Kitoh, A.: APHRODITE: Constructing a Long-Term Daily Gridded Precipitation Dataset for Asia Based on a Dense Network of Rain Gauges, B. Am. Meteorol. Soc., 93, 1401–1415, doi:10.1175/BAMS-D-11-00122.1, 2012.

2. Table 2: What is the basis of sensitivity rank of model parameters? Please provide the reference of the Sensitivity Rank (column-3), Default Value (column-4) and Range (column-5).

---

## Referee Comment (RC2) · Anonymous Referee #2 · 23 Apr 2017

The manuscript entitled "Hydrologic modeling of a Himalayan mountain basin by using the SWAT model" presents an application of a hydrologic model to simulate historical streamflow in a topographically complex and data scarce Himalayan mountain system. The authors are putting their great efforts in this study. This type of research issue can help for making better informed decisions regarding future water management strategies of the Himalayan mountain regions. However, I have some critical comments as given below, and want the authors to address thoroughly before considering this manuscript for further processes.

My major comments are: (1) I don't find a concrete innovation in modeling technique to publish in this journal. The introduction section is too lengthy including very general statements and it seems like a review paper. This should be concise based on the overall study objectives. (2) The methods section should be revised thoroughly

since it contains the equations which are explained in details in SWAT documentation and in several previous research papers. The abstract section says that the authors have used manual calibration approach but as I understand from methods section, they have explained the application of SWAT-CUP SUFI-2 approach for model calibration. This is very confusing to the audience. (3) One of my critical comments is regarding the study of uncertainty analysis in this paper. In Abstract section, the authors have stated that they have done quantification of uncertainty analysis but not mentioned in details in the text. This is very important aspect of model simulation studies so this should be accounted very properly in the paper. (4) The paper is lacking supporting references in many places. It also contains several technical errors. Specific suggestions about the paper are listed below: Title: Since the study is primarily focused for calibration and validation of Ganga River Basin SWAT model, the Title should be modified "should use different terms instead of using hydrologic modeling". Abstract section, lines 10-13: I am not convinced with these statements. There are several studies conducted in several Himalaya Mountain basins. I suggest the authors to see a few of the following examples: 1. Neupane, R. P., Yao, J., & White, J. D. (2014). Estimating the effects of climate change on the intensification of monsoonal‐driven stream discharge in a Himalayan watershed. Hydrological Processes, 28(26), 6236-6250. 2. Neupane, R. P., White, J. D., & Alexander, S. E. (2015). Projected hydrologic changes in monsoon-dominated Himalaya Mountain basins with changing climate and deforestation. Journal of Hydrology, 525, 216-230. 3. Nepal, S., & Shrestha, A. B. (2015). Impact of climate change on the hydrological regime of the Indus, Ganges and Brahmaputra river basins: a review of the literature. International Journal of Water Resources Development, 31(2), 201-218. Abstract section, line 16: Should be clear in manual and automatic calibration approaches in the paper. Abstract section, line 20: "between 13-20%" > should have clear explanation of estimating these numbers in the text. Introduction section: The first sentence is not so clear. Introduction section, lines 28-33: Authors should provide proper references to support these statements. Introduction section, 2nd paragraph: The first sentence is not clear and should be

re-written. Introduction section, lines 50-52: Provide the reason of getting better NS value in monthly basis with reference. Introduction section, paragraphs 3 and 4: These paragraphs mostly include literature reviews regarding water quality issues. This is not appropriate since the paper is focused for water quantity issues. So, these should be removed. Introduction section, lines 106-107: provide supporting reference for this data. Authors should trim the introduction section based on research objectives. Page 4, line 143: "computes" > compute. Page 7, line 236: "km2and" > km2 and Page 7, line 236: "7785 m" > 7785 m mean above sea level (masl) (should be consistent through-out the paper). Page 7, lines 244-248: These are very important information of the basin and should have supporting references. Page 7, lines 249-250: Why these two sentences are separate? can be merged with below paragraph. Page 8, lines 276-277: This sentence should be merged with below paragraph. Page 8, line 288: ".To obtain" > Should have space after full stop (.) (this should be corrected throughout the paper. Page 8, lines 288-290: What is the basis for these threshold values? Page 8, lines 294-295: The sentence is not correct and what is the basis of selecting hydrologic pa-rameters for this study? Page 9, line 325: "was" > were Conclusions section: The first sentence is not complete, using what? Table 1: I suggest to include the dates of data derived to use for land use, streamflow, and weather information. Table 3: This should be re-constructed separating calibration and validation sections properly. Figure 1: The caption is not complete. This should explain every details of the figure and should be followed for all the figures. The "Location of what?" in the Legend. Figures 4 and 5 can be merged in Figure 1. In summary, the paper has both scientific and technical flaws as mentioned above, so I would ask authors for a comprehensive revision based on the above-mentioned comments to better improve from the present form.

---

## Short Comment (SC1) · 23 Apr 2017

Just a very brief comment. The study location has a high degree of seasonality, so presumably the NSE values for monthly data were calculated using the discharge means for each individual month, otherwise the values will be artificially high. Perhaps the paper could include the expression used to calculate the NSE values for the monthly discharge data.

---

## Referee Comment (RC3) · Anonymous Referee #3 · 26 Apr 2017

General comments

The manuscript 'Hydrologic modeling of a Himalaya mountain basin by using the SWAT model' it used in the northwestern Himalaya which is part of the larger Ganga River Basin. General comments for this paper are summarized below. 1. Although the paper addresses an issues of importance regarding the need for hydrological studies in the NW Himalaya, the paper in itself does not address any relevant scientific questions. 2. As a result, the paper lacks novelty and simply appears as a hydrological modeling exercise for this region. 3. Most of the paper also focuses on the model details and parameterization rather than any scientific question. 4. The paper also lacks references

to major studies conducted in the Himalayan region that have advanced our understanding of the hydro-climatology of the region. (see below in specific comments) 5. I suggest the authors to go back and formulate specific scientific questions they want to address regarding the high-altitude hydrology in the NW Himalaya. As it stands, the paper needs a complete revision and would suggest the authors to perhaps resubmit as a new submission.

Specific comments

1. The title of the paper is too generic.

2. The first paragraph in the introduction section is lacking any citation or reference.

3. Line 116. At the same time, there has been a number of studies that have used spatially distributed hydrological models in the context of Himalayan regions to simulate streamflow (Immerzeel et al., 2013; Lutz et al, 2013).

Lutz, A.F., Immerzeel, W.W., Shrestha, A.B. and Bierkens, M.F.P., 2014. Consistent increase in High Asia's runoff due to increasing glacier melt and precipitation.ÂăNature Climate Change,Âă4(7), pp.587-592

Immerzeel, W.W., Pellicciotti, F. and Bierkens, M.F.P., 2013. Rising river flows throughout the twenty-first century in two Himalayan glacierized watersheds.ÂăNature geoscience,Âă6(9), pp.742-745.

4. Line 255. APHRODITE tends to underestimate high-altitude precipitation. It might be important to use station data where appropriate and valid. Underestimation of monsoon in Figures 9 and 10 are probably as a result of using APHRODITE data.

5. A majority of the results and discussion is spent on calibration/validation and model parameterization and performance. Only the last paragraph discusses some of the model outcome which I find the most interesting part of the entire study. Further discussion detailing these results would shed more light on perhaps the more interesting scientific questions of this study.

6. I am not certain if Figures 2-4 are completely necessary by themselves are necessary. These can be combined into one figures.

7. Figure 9 compares the average monthly simulated and observed streamflow. Why not plot the observed and simulated monthly streamflow for both the calibration and validation period?

---

## Author Comment (AC1) · 23 Jun 2017

OK. We would include the expression for NSE as well as for the coefficient of determination.

---

## Author Comment (AC2) · 29 Jun 2017

The response is given in the supplemental file (PDF)

Please also note the supplement to this comment:
https://www.hydrol-earth-syst-sci-discuss.net/hess-2017-100/hess-2017-100-AC2-supplement.pdf
* * *

---

## Author Comment (AC3) · 29 Jun 2017

The response is given in supplement file

Please also note the supplement to this comment:
https://www.hydrol-earth-syst-sci-discuss.net/hess-2017-100/hess-2017-100-AC3-supplement.pdf